# Involvement of Antioxidant and Prevention of Mitochondrial Dysfunction, Anti-Neuroinflammatory Effect and Anti-Apoptotic Effect: Betaine Ameliorates Haloperidol-Induced Orofacial Dyskinesia in Rats

**DOI:** 10.3390/brainsci13071064

**Published:** 2023-07-12

**Authors:** Hsiang-Chien Tseng, Mao-Hsien Wang, Chih-Hsiang Fang, Yi-Wen Lin, Hung-Sheng Soung

**Affiliations:** 1Department of Anesthesiology, Shin Kong Wu Ho-Su Memorial Hospital, Taipei 11101, Taiwan; 2School of Medicine, Fu Jen Catholic University, New Taipei City 24205, Taiwan; 3Department of Anesthesia, En Chu Kon Hospital, Sanshia District, New Taipei City 23702, Taiwan; 4China Medical University Hospital, Taichung 404332, Taiwan; 5Trauma and Emergency Center, China Medical University Hospital, Taichung 404018, Taiwan; 6Institute of Biomedical Engineering, National Taiwan University, Taipei 10051, Taiwan; 7Department of Psychiatry, Yuan-Shan Branch of Taipei Veteran General Hospital, Yilan 26604, Taiwan; 8Department of Biomedical Engineering, National Defense Medical Center, Taipei 11490, Taiwan

**Keywords:** antioxidant, betaine, haloperidol, mitochondrial dysfunction, orofacial dyskinesia

## Abstract

With its pathophysiological characteristics strongly similar to patients with tardive dyskinesia (TD), haloperidol (HP)-induced neurotoxicity and orofacial dyskinesia (OD) in animal models have long been used to study human TD. This study aimed to explore the potential protective effects of betaine (BT), a vital biochemical compound present in plants, microorganisms, animals, and various dietary sources. The study focused on investigating the impact of BT on haloperidol (HP)-induced orofacial dyskinesia (OD) in rats, as well as the underlying neuroprotective mechanisms. To induce the development of OD, which is characterized by increased vacuous chewing movement (VCM) and tongue protrusion (TP), rats were administered HP (1 mg/kg i.p.) for 21 consecutive days. BT was administered intraperitoneally (i.p.) at doses of 30 and 100 mg/kg, 60 min later, for 21 successive days. On the 21st day, after evaluating OD behavior, the rats were sacrificed, and various measurements were taken to assess the nitrosative and oxidative status, antioxidant capacity, mitochondrial function, neuroinflammation, and apoptotic markers in the striatum. The results demonstrated that (1) HP induced OD development, and (2) BT was found to prevent most of the HP-induced OD; decrease oxidative stress levels; increase anti-oxidation power; prevent mitochondrial dysfunction; and reduce the levels of neuroinflammatory and apoptotic markers in the striatum. Our results demonstrate that the neuroprotective effects of BT against HP-induced OD are credited to its antioxidant prevention of mitochondrial dysfunction, anti-neuroinflammatory effects, and anti-apoptotic effects, suggesting that BT may be a novel therapeutic candidate in delaying or treating human TD in clinical settings. However, further studies will be warranted to extrapolate preclinical findings into clinical studies for a better understanding of the role of BT.

## 1. Introduction

Tardive dyskinesia (TD) is a hyperkinetic movement disorder that affects the orofacial region and is caused by prolonged use of neuroleptic drugs, particularly first-generation antipsychotics. It is characterized by involuntary movements, such as choreiform, athetoid, and rhythmic motions, predominantly involving the mouth, face, and tongue [1]. TD is considered a significant clinical concern in the treatment of schizophrenia because it can persist even after discontinuation of the medication and, in around 2% of patients, it is irreversible [1,2]. Haloperidol (HP), a first-generation antipsychotic belonging to the butyrophenone class, is commonly used to manage agitation and aggressiveness during the acute phase of schizophrenia. In animal models, prolonged treatment with HP for 21 consecutive days leads to a blockade of dopamine (DA) D2 receptors, promoting side effects and resulting in neurotoxicity and the development of orofacial dyskinesia (OD). OD is characterized by increased vacuous chewing movement (VCM), tongue protrusion (TP), and orofacial burst [3]. This model has been extensively used to study the neuropathology of TD and behavioral symptoms similar to Parkinson’s disease (PD) [3]. The development of HP-induced OD in rats only and mimicking TD observed in humans, has been associated with elevated production of nitric oxide (NO) and byproducts of lipid peroxidation (LPO) in the striatum, along with increased levels of inflammatory mediators. Additionally, there is a significant decrease in the activity of antioxidant enzymes and neurodegeneration in the striatum, resulting in motor dysfunction resembling PD symptoms [4,5,6,7,8,9,10,11,12]. Although the precise mechanisms underlying the pathophysiology of OD are not fully understood, previous studies have suggested that treatment with antioxidant or anti-inflammatory agents may have potential therapeutic effects on TD in animal models [5,6,7,10,11,12].

No treatment for TD is available in clinical practice; hence, there is a need to explore new therapeutic approaches for TD in medical research [1,2]. Betaine (N,N,N-trimethylglycine; BT), also known as glycine betaine, found from Beta vulgaris (beet), is a zwitterionic amino acid derivative that is abundantly present in various food sources like wheat bran, spinach, and seafood [13,14]. It is a non-toxic and safe dietary nutrient supplement widely used in both humans and animals [13,14]. Upon ingestion, BT is rapidly absorbed from the digestive tract, primarily through proton-coupled amino acid transport and sodium-dependent amino acid transport systems. It is subsequently distributed to different organs, including the brain [15,16]. BT exhibits various pharmacological effects, including antioxidant and anti-inflammatory activities, as well as the ability to mitigate endoplasmic reticulum stress and apoptosis [13,15,17,18,19,20,21,22]. Due to these properties, it is anticipated that BT may possess neuroprotective characteristics suitable for addressing behavioral impairments induced by ischemia, toxins, and stress, such as seizures, depressive-like symptoms, motor dysfunction, and memory deficits [22,23,24,25,26,27]. However, despite extensive research on the beneficial effects of BT, there is currently a dearth of knowledge regarding its potential therapeutic use as a neuroprotectant against HP-induced OD and the underlying mechanisms involved in its ameliorative effects. Thus, investigating the beneficial effects of BT in this context is crucial and could provide valuable insights into its therapeutic potential.

Hence, the hypothesis of this study postulates that BT possesses the capacity to counteract the nitrosative and oxidative damage, mitochondrial dysfunction, neuroinflammation, and apoptosis induced by HP in the striatum. Consequently, it is expected that BT administration can subsequently impede the development of OD in animals. This hypothesis is based on the well-documented and robust anti-oxidative, anti-inflammatory, and anti-apoptotic properties exhibited by BT in previous studies conducted by our laboratory and others [13,15,17,18,19,20,21,22,24,26,27]. The purpose of the present study was designed to explore the possible effects of BT on HP-induced OD using a well-established rat model of neurotoxicity. This study examined: (1) behaviors linked to OD; (2) the nitrosative and oxidative status; (3) antioxidant power; (4) mitochondrial function; (5) neuroinflammation; and (6) caspase-3 in the striatum of rats, since it has been previously reported that altered striatal nitrosative and oxidative stress, neuroinflammation, mitochondrial dysfunction, and severe neurodegeneration were tightly correlated with HP-induced OD in animal models, mimicking TD observed in clinical settings [4,5,6,7,8,9,10,11,12,28].

## 2. Materials and Methods

### 2.1. Animals

All experimental procedures conducted in this study adhered to the “Guidelines for the Care and Use of Laboratory Animals” published by the U.S. National Institutes of Health. Additionally, the Institutional Animal Care and Use Committee (IACUC) of National Taiwan University College of Medicine approved all experimental protocols (IACUC Approval No: 20190605). In order to exclude that the menstrual cycle of female rats may affect the experimental results, male rats were selected as experimental animals in this experiment. Male Wistar rats weighing between 270 and 320 g, approximately three months old, were obtained from BioLASCO Taiwan Co., Ltd., Taipei, Taiwan. The rats were housed in groups of three in Plexiglas cages and provided with ad libitum access to food and water. The animal housing room maintained a controlled temperature of 22 ± 3 °C and a 12/12 h light/dark cycle, with lights turned on at 7:00 a.m. To minimize animal anxiety and discomfort, a gentle handling procedure was implemented prior to the commencement of the experimental procedures. Each rat was handled for 20 min per day, for a duration of four days, in order to acclimate them to human interaction and reduce any potential stress.

### 2.2. Drugs

Haloperidol (HP) and betaine (BT) used in this study were obtained from Sigma, located in St. Louis, Missouri, USA. HP is identified as 4-[4-(4-Chlorophenyl)-4-hydroxy-1-piperidinyl]-1-(4-fluorophenyl)-1-butanone, with a CAS number of 52-86-8. BT is known as N,N,N-trimethylglycine, with a CAS number of 107-43-7. In order for the drug concentration to enter the animal body accurately, HP and BT were prepared using normal saline (0.9% NaCl) as the solvent and administered intraperitoneally (i.p.) on a daily basis for a period of 21 days. Fresh solutions of the drugs were prepared prior to each administration. The dosages of the drugs used in this study were determined based on the previously published literature [5,11,12,23]. BT was administered and pretested at a low dose of 1 mg/kg, owing to no evident effect for the low-dosage application; the dose was increased gradually and eventually up to 300 mg/kg to observe the statistically significant effect for BT treatment. Therefore the 30/100 mg/kg dosage was adapted to the current study. HP and BT were administered at a volume of 2.0 mL/kg of body weight.

### 2.3. Experimental Groups and Drug Treatment

The rats were randomly assigned to six groups, each group having eight rats:HP treatment group (HP): HP (1 mg/kg i.p.) for 21 days;HP + BT 30 mg/kg treatment group (HP + BT30): HP (1 mg/kg i.p.) + BT (30 mg/kg, i.p.) for 21 days;HP + BT 100 mg/kg treatment group (HP + BT100): HP (1 mg/kg i.p.) + BT (100 mg/kg, i.p.) for 21 days;BT 30 mg/kg treatment group (BT30): BT (30 mg/kg, i.p.) for 21 days;BT 100 mg/kg treatment group (BT100): BT (100 mg/kg, i.p.) for 21 dayslControl group (C): Normal saline (i.p.) for 21 days.

HP (1 mg/kg, i.p.) was administered for 21 successive days to induced OD. BT (30 or 100 mg/kg), respective to the assigned group, was administered i.p. 60 min after injecting HP for 21 successive days. Figure 1 illustrates the experimental paradigm.

### 2.4. Behavioral Assessment of Orofacial Dyskinesia

The assessment of OD behavior in the rats was conducted according to established protocols in our laboratory, as reported previously [7,12]. The evaluations were performed at 6 h after the administration of HP and BT or normal saline (C group) on days 1, 7, 14, and 21 of the study. To ensure objectivity and eliminate any potential bias, each rat was assigned a unique identification number, and all behavioral assessments were carried out independently by two experienced researchers. These researchers were unaware of the treatment groups and remained blinded to all experimental conditions throughout the study. They were not informed about which rats received specific treatments or belonged to particular groups. For the assessment of OD behavior, each rat was placed individually in a specially designed assessment cage measuring 20 cm × 20 cm × 19 cm. The floor of the cage was equipped with mirrors, allowing for behavioral quantification even when the rat was facing away from the observers. The occurrence of OD, as well as the frequency of vacuous chewing movement (VCM) and tongue protrusion (TP) events, were recorded for a duration of 5 min in each session following a 2-min adaptation period. All behavioral experiments were consistently conducted between 09:00 a.m. and 11:00 a.m.

### 2.5. Biochemical Measurement

On the 21st day of the study, the rats were sacrificed approximately 1 h after the behavioral quantification. The brains were swiftly removed and rinsed with ice-cold isotonic saline solution to eliminate any residual blood. The brain was put on ice; the striatum, a specific brain region, was carefully dissected from the intact brain on an ice plate, following the stereotaxic atlas provided by Budantsev et al. [29]. The dissected striatum tissue was rinsed with isotonic saline and weighed. It was then homogenized in 0.1 N HCl. A 10% (*w*/*v*) tissue homogenate was prepared using a 0.1-M phosphate buffer with a pH of 7.4. For the catalase (CAT) assay, the post-nuclear fraction was obtained by subjecting the homogenate to centrifugation at 1000× *g* for 20 min at a temperature of 4 °C. For the remaining enzyme assays, the homogenate was centrifuged at 12,000× *g* for 60 min at 4 °C.

### 2.6. Determination of Nitrites Concentration

The level of nitrites, which are the final products of nitric oxide (NO) metabolism, was determined using Roche’s “NO colorimetric assay” [30]. This method involves the reaction between nitrite–sulfanilamide and N-(1-naphthyl)-ethylenediamine dihydrochloride, resulting in the formation of a reddish-violet diazo dye. The measurement of this dye’s absorbance is performed spectrophotometrically at a wavelength of 540 nm. To determine the nitrite level in the tissue, a mixture of 100 μL of the tissue homogenate and 400 μL of redistilled water was incubated in a hot water bath at 100 °C for 15 min to halt all enzymatic processes. After cooling, 30 μL of Carrez I reagent (0.36 M K_4_[Fe(CN)_6_] × 3 H_2_O) and 30 μL of Carrez II reagent (1 M ZnSO_4_ × 7 H_2_O) were added to each sample. The samples were then alkalized to pH 8.0 by adding 4 μL of 10 M NaOH (Sigma, St. Louis, MI, USA ) and centrifuging at 10,000× *g* before further use. For nitrite determination, 75 μL of the supernatant and 75 μL of redistilled water were added to each well of a microplate. In the blank samples, redistilled water was used instead of the supernatant. The samples were incubated at 25 °C for 30 min, and the absorbance was measured at 540 nm. Subsequently, the tested samples were subjected to color development by adding 50 μL of a 1% solution of sulfanilamide in 2.5% H_3_PO_4_ and 50 μL of a 0.1% solution of N-(1-naphthyl)-ethylenediamine dihydrochloride in 2.5% H_3_PO_4_. After thorough mixing, the microplates were kept in the dark for 15 min, and the absorbance was measured again at 540 nm. The results were calculated based on the standard curves derived from different concentrations of sodium nitrite solutions (ranging from 6 to 600 μM), using the change in absorbance measured before and after incubation with sulfanilamide and N-(1-naphthyl)-ethylenediamine dihydrochloride. The nitrite concentration, expressed in μg/mL, served as an index for the NO level in the tested tissue sample.

### 2.7. Assessment of Lipid Peroxidative Indices

Lipid peroxide concentration was measured by the thiobarbituric acid reactive substance (TBARS) assay adapted from Ohkawa et al. [31]. According to the procedures described by Hashimoto et al. [24], the concentration was measured in nmol malondialdehyde/mg protein. Malondialdehyde levels were then further normalized to a standard preparation of 1,1,3,3-tetra ethoxypropane.

### 2.8. Measurement of Glutathione

GSH levels were determined using Ellman’s method [32]. To the homogenate, 10% trichloroacetic acid was added. The mixture was centrifuged at 8000× *g* for 10 min, and subsequently, 1.0 mL of Ellman’s reagent (19.8 mg of 5, 5-0-dithiobisnitro benzoic acid in 100 mL of 1.0% sodium citrate and 3 mL of a phosphate buffer [pH 8.0]) were added. The absorbance of the final product was measured at 412 nm. The results were expressed in nmol GSH/mg tissue.

### 2.9. Measurement of Superoxide Dismutase Activity

The assay used to determine if the superoxide dismutase (SOD) activity was based on the ability of SOD to inhibit the spontaneous oxidation of adrenaline to adrenochrome [33]. To conduct the assay, 0.05 mL of the supernatant was mixed with 2.0 mL of a carbonate buffer and 0.5 mL of ethylenediaminetetraacetic acid (EDTA). The reaction was initiated by adding 0.5 mL of epinephrine, and the auto-oxidation of adrenaline (at a concentration of 3 × 10^−4^ M) to adrenochrome was measured at an optical density of 480 nm. The changes in optical density were measured every minute at 480 nm and normalized to a blank reagent. The results were expressed as units of SOD activity per milligram of protein, with one unit of SOD activity defined as the amount that induced approximately 50% inhibition of adrenaline oxidation. The results were further expressed as nanomoles of SOD per unit/milligram of tissue.

### 2.10. Measurement of Catalase Activity

The catalase (CAT) activity assay was adapted from Beers and Sizer [34]. The reaction mixture contained 2 mL of a phosphate buffer (pH 7.0), 0.95 mL of hydrogen peroxide (0.019 M), and 0.05 mL of the supernatant, making a total volume of 3 mL. The absorbance of the reaction mixture was recorded at 240 nm every 10 s for 1 min. One unit of CAT activity was defined as the amount of enzymes required to decompose 1 millimole of hydrogen peroxide per minute at 25 °C and pH 7.0. The results of the CAT activity were expressed as units per milligram of protein. The units of activity were determined from a standard graph of hydrogen peroxide. The final results were reported as CAT activity per unit/milligram of tissue.

### 2.11. Measurement of Mitochondrial Function

Briefly, a 10% homogenate of the striatum was prepared in an ice-cold Tris-Sucrose buffer (0.25 M, pH 7.4) using a glass Teflon grinder at 4 °C. The homogenate was centrifuged at 1000× *g* for 10 min at 4 °C to obtain the nuclear pellet. The supernatant was further subjected to centrifugation at 10,000× *g* for 20 min at 4 °C to obtain the mitochondrial pellet and cytosol. The pellet was washed three times in a Mannitol–Sucrose–HEPES buffer (pH 7.4) and resuspended in the same buffer. The activity of succinate dehydrogenase (SDH) was determined as described by Pennington [35], with minor modifications. Briefly, mitochondrial protein (0.05 mg) was incubated with 50 mM of potassium phosphate (pH 7.4) containing sodium succinate (0.01 mol/L) and p-iodonitrotetrazolium violet (2.5 lg/mL) for 10 min. The reaction was stopped by the addition of 10% TCA. The color obtained was extracted with ethyl acetate: ethanol: trichloroacetic acid (5:5:1, *v*:*v*:*w*) and measured at 490 nm. The activity was expressed as an OD value at 490 nm/mg protein. Total ATPase activity was measured by determining the inorganic phosphate liberated from ATP as described by Prasad and Muralidhara [36]. The reaction was initiated by the addition of cytosolic protein (50 lg) to a reaction mixture containing a Tris HCl buffer (0.02 M, pH 7.4), NaCl (100 mM), KCl (20 mM), and MgCl2 (5 mM) and was incubated for 15 min at 37 °C. The reaction was arrested by adding 20% TCA. The mixture was centrifuged (15,009× *g*; 10 min) and the phosphate in the protein-free supernatant was estimated. Blanks without enzymes were used throughout the entire procedure. Enzyme activity was expressed as the lg of the liberated inorganic phosphate/mg protein. NADH-cytochrome C reductase (complex I–III) and succinate-cytochrome C reductase (complex II–III) were determined following standard methods [37].

### 2.12. Measurement of Neuroinflammatory Markers

The quantification of TNF-α, IL-1β, and IL-6 was performed using an immunoassay kit from KRISHGEN BioSystem, located in Ashley Ct, Whittier, CA. The specific kit used was the quantikine rat TNF-α, IL-1β, and IL-6 immunoassay, which is a solid-phase, sandwich enzyme-linked immunosorbent assay (ELISA) designed to measure the levels of these cytokines in rat samples. The assay involved a 4.5 h procedure using a microtiter plate reader. Concentrations of TNF-α, IL-1β, and IL-6 were determined by comparing the absorbance values of the samples to the standard curves generated with known concentrations of the respective cytokines. The results were expressed in picograms per milliliter (pg/mL) of protein units.

### 2.13. Measurement of the Apoptotic Marker Caspase-3

Caspase-3, also known as CPP-32, Yama, or Apopain, is an intracellular cysteine protease involved in apoptosis. In its inactive form, caspase-3 exists as a pro-enzyme. However, during the cascade of events linked to apoptosis, it is activated. The protease activity of caspase-3 in tissue lysates or homogenates can be evaluated using a caspase-specific peptide conjugated to the color reporter molecule p-nitroaniline (pNA). When caspase cleaves the peptide, it releases the chromophore pNA, which can be measured spectrophotometrically at 405 nm. The color reaction directly correlates with caspase enzyme activity in the cell lysate or homogenate. For the assessment of caspase activity, the R&D Systems caspase-3 colorimetric kit (Cat No. GTX85558, GeneTex Inc., Hsinchu, Taiwan) was utilized. This kit provides the necessary reagents to measure caspase-3 activity. The results of the assay were expressed as nanomoles of pNA released per milligram of protein, reflecting the caspase-3 enzymatic activity in the sample.

### 2.14. Determination of Protein Content

The cytosolic and mitochondrial protein content values were determined using bovine serum albumin (obtained from Sigma, St. Louis, MI, USA) as a standard [38].

### 2.15. Statistical Analysis

All data were analyzed using GraphPad Prism 8.3.0 (GraphPad Software Inc., San Diego, CA, USA). The behavioral assessment data were analyzed using two-way analysis of variance (ANOVA); however, the biochemical estimations were analyzed using one-way ANOVA. Post hoc comparisons between groups were performed using Tukey’s test. A *p*-value of less than 0.05 was considered statistically significant to indicate significant differences between groups. The data were expressed as mean ± standard error of the mean (SEM).

## 3. Results

Before HP administration, the values of the tested parameters were very similar for the different groups. Tukey’s test parameters revealed no significant differences between the group treated with BT (30 or 100 mg/kg; BT30, or BT100) and the C groups (BT30 vs. C, *p* > 0.05; BT100 vs. C, *p* > 0.05).

### 3.1. Effect of BT on the HP-Induced Increases in the Frequency of VCM and TP

HP induced significant increases in the VCM (Figure 2A) and TP (Figure 2B) on the 1st, 7th, 14th, and 21st day. As compared to the C group, the HP group showed higher VCM and TP on the 1st day (VCM: from 1.23 ± 0.2 to 16.5 ± 3.6 counts/5 min, *p* < 0.001 and TP: from 2.1 ± 0.8 to 7.9 ± 1.1 counts/5 min, *p* < 0.001), 7th day (VCM: from 1.12 ± 0.3 to 25.4 ± 3.9 counts/5 min, *p* < 0.001 and TP: from 1.74 ± 0.3 to 10.5 ± 1.1 counts/5 min, *p* < 0.001), 14th day (VCM: from 0.9 ± 0.2 to 35.7 ± 4.2 counts/5 min, *p* < 0.001 and TP: from 1.86 ± 0.5 to 17.2 ± 2.8 counts/5 min, *p* < 0.001), and 21st day (VCM: from 1.15 ± 0.7 to 54.8 ± 4.7 counts/5 min, *p* < 0.001 and TP: from 1.4 ± 0.6 to 25.9 ± 2.9 counts/5 min, *p* < 0.001). It was observed that 30 and 100 mg/kg of BT did not have a significant effect on the increased VCM and TP induced by HP on the 1st and 7th day (*p* > 0.05). However, on the 21st day, 30 mg/kg of BT reduced VCM and TP (VCM: by −22.26% from 54.8 ± 4.7 to 42.6 ± 4.5 counts/5 min, *p* < 0.001 and TP: by −29.34% from 25.9 ± 2.9 to 18.3 ± 1.6 counts/5 min, *p* < 0.001). It was found that 100 mg/kg of BT reduced VCM and TP on the 14th day (VCM: by −25.4% from 35.7 ± 4.2 to 26.6 ± 4.7 counts/5 min, *p* < 0.001 and TP: by −16.86% from 17.2 ± 2.8 to 14.3 ± 2.7 counts/5 min, *p* < 0.001) and on the 21st day (VCM: by −48.51% from 54.8 ± 4.7 to 36.9 ± 4.4 counts/5 min, *p* < 0.001 and TP: by −39.38% from 25.9 ± 2.9 to 15.7 ± 2.2 counts/5 min, *p* < 0.001), respectively. These results indicate that the BT treatment prevented HP-induced OD.

### 3.2. Effect of BT on the HP-Induced Increases in Striatal Nitric Oxide and Lipid Peroxide Production

HP induced a significant increase in the striatal levels of NO (nitrites) from 110.43 ± 9.64 to 266.14 ± 12.8 μg/mL (*p* < 0.001) (Figure 3A), and TBARS from 30.7 ± 3.2 to 61.71 ± 4.3 nmol/mg protein) (*p* < 0.001) (Figure 3B) in the HP groups on the 21st day. The increases in the nitrite and TBARS levels in the HP groups were significantly inhibited by 30 mg/kg of BT (nitrites: by −28.72% from 266.14 ± 12.8 to 189.7 ± 10.8 μg/mL, *p* < 0.001 and TBARS: by −27.78% from 61.71 ± 4.3 to 44.57 ± 3.36 nmol/mg protein, *p* < 0.001) and 100 mg/kg (nitrites: by −42.83% 266.14 ± 12.8 to 152.14 ± 9.5 μg/mL, *p* < 0.001 and TBARS: by −39.12% from 61.71 ± 4.3 to 37.57 ± 2.5 nmol/mg protein, *p* < 0.001), respectively. These results suggest that the BT treatment could effectively inhibit HP-induced increases in striatal nitric oxide and lipid peroxide production.

### 3.3. Effect of BT on the HP-Induced Decreases in Striatal Antioxidation Power

As compared with the C group, the striatal levels of GSH (Figure 4A) and protective enzymes such as SOD (Figure 4B) and CAT (Figure 4C) were significantly decreased (GSH: from 14.82 ± 1.19 to 5.78 ± 0.78 nmol/mg tissue, *p* < 0.001; SOD: from 2.81 ± 0.23 to 1.48 ± 0.13 U/mg tissue, *p* < 0.001 and CAT: from 8.25 ± 0.73 to 2.12 ± 0.42 U/mg tissue, *p* < 0.001) in the HP groups on the 21st day. In addition, these reductions in the levels of GSH, SOD, and CAT in the HP groups were also significantly attenuated by 30 mg/kg of BT (GSH: by 34.51% to 8.9 ± 0.63 nmol/mg tissue, *p* < 0.001; SOD: by 48.87% to 2.13 ± 0.23 U/mg tissue, *p* < 0.001 and CAT: by 45.19% to 4.89 ± 0.71 U/mg tissue, *p* < 0.001) and 100 mg/kg of BT (GSH: by 63.27% to 11.5 ± 1.29 nmol/mg tissue, *p* < 0.001; SOD: by 72.93% to 2.45 ± 0.13 U/mg tissue, *p* < 0.001 and CAT: by 69% to 6.35 ± 0.72 U/mg tissue, *p* < 0.001) according to a two-way ANOVA and post hoc analysis. Therefore, the results suggest that the BT treatment prevented HP-induced decreases in striatal antioxidation power.

### 3.4. Effect of BT on the HP-Induced Striatal Mitochondrial Dysfunction

The striatal levels of SDH (Figure 5A), total ATPase (Figure 5B), NADH-cytochrome C reductase (Figure 5C), and succinatecytochrome C reductase (Figure 5D) were significantly decreased (SDH: from 10.9 ± 1 to 5.7 ± 0.6 OD at 490 nm/mg protein, *p* < 0.001; total ATPase: from from 290.5 ± 10.4 to 205.8 ± 14.9 μg Pi released/mg protein, *p* < 0.001; NADH-cytochrome C reductase: from 32.7 ± 3.2 to 20.7 ± 2.7 nmol cyt C reduced/min/mg protein, *p* < 0.001 and succinatecytochrome C reductase: from 11.5 ± 0.8 to 4.9 ± 0.8 nmol cyt C reduced/min/mg protein, *p* < 0.001) in the HP groups on the 21st day. The diminished levels of SDH, total ATPase, NADH cytochrome C reductase, and succinate-cytochrome C reductase in the H groups were also elevated by 30 mg/kg of BT (SDH: 7.3 ± 0.6 OD at 490 nm/mg protein, *p* < 0.001; total ATPase: 251.4 ± 10.1 μg Pi released/mg protein, *p* < 0.001; NADH-cytochrome C reductase: 26.7 ± 2.7 nmol cyt C reduced/min/mg protein, *p* < 0.001 and succinate-cytochrome C reductase: 7.7 ± 0.6 nmol cyt C reduced/min/mg protein, *p* < 0.001) and the 100 mg/kg BT treatment (SDH: 8.8 ± 0.9 OD at 490 nm/mg protein, *p* < 0.001; total ATPase: 264.6 ± 15.4 μg Pi released/mg protein, *p* < 0.001; NADH-cytochrome C reductase: 29.7 ± 2.6 nmol cyt C reduced/min/mg protein, *p* < 0.001 and succinate-cytochrome C reductase: 9.3 ± 0.7 nmol cyt C reduced/min/mg protein, *p* < 0.001). These results indicate that the BT treatment prevented HP-induced striatal mitochondrial dysfunction.

### 3.5. Effect of BT on the HP-Induced Increases in Striatal Neuroinflammatory and Apoptotic Markers

According to Figure 6, striatal levels of TNF-α (Figure 6A), IL-1β (Figure 6B), IL-6 (Figure 6C), and caspase-3 (Figure 6D) were significantly increased (TNF-α: from 41.7 ± 3.9 to 119.1 ± 7.4 pg/mL protein, *p* < 0.001; IL-1β: from 36.7 ± 2.9 to 106.1 ± 7.6 pg/mL protein, *p* < 0.001; IL-6: from 40.7 ± 2.43 to 119.2 ± 12.43 pg/mL protein, *p* < 0.001 and caspase-3: from 1.8 ± 0.36 to 4.89 ± 0.4 nmol/mg protein, *p* < 0.001) after HP treatment for 21 successive days in the HP groups. As expected, these increases in TNF-α, IL-1β, IL-6, and caspase-3 levels in the HP-treated animals were significantly inhibited by the 30 mg/kg BT treatment (TNF-α: 78.86 ± 6.3 pg/mL protein, *p* < 0.001; IL-1β: 72.71 ± 7 pg/mL protein, *p* < 0.001; IL-6: 75.57 ± 4.24 pg/mL protein, *p* < 0.001 and caspase-3: 3.44 ± 0.38 nmol/mg protein, *p* < 0.001) and 100 mg/kg BT treatment (TNF-α: 63.7 ± 5.2 pg/mL protein, *p* < 0.001; IL-1β: 50.3 ± 4.7 pg/mL protein, *p* < 0.001; IL-6: 61.71 ± 6.5 pg/mL protein, *p* < 0.001 and caspase-3: 2.37 ± 0.29 nmol/mg protein, *p* < 0.001). These results indicate that the BT treatment inhibited HP-induced increases in sciatic neuroinflammatory activity as well as apoptosis markers.

## 4. Discussion

In our research, we have emphasized the beneficial effects of BT in guarding against involuntary HP-induced OD, nitrosative and oxidative harm, mitochondrial dysfunction, neuroinflammation, and activation of the apoptotic pathway in animal experiments. Based on our current understanding, our findings reveal that BT exhibits protective properties against HP-induced pathophysiological dysfunction, suggesting its potential therapeutic value in future clinical studies. Not only do the characteristic features of HP-induced OD bear a strong resemblance to the symptoms of TD, but HP has also been implicated in the development of TD in humans. In rats, HP-induced changes in VCM (vacuous chewing movements) and TP (tongue protrusions) have proven to be a valuable model and an essential tool for identifying potential therapeutic agents for TD [3,4]. However, it is important to acknowledge that there are numerous concerns and debates surrounding the use of animal models for studying TD. Our findings regarding OD induced by HP align with those of other researchers and our own previous studies [3,4,5,6,7,10,11,12]. In our experiments, rats treated with HP for 21 consecutive days exhibited a significant increase in the frequency of VCM and TP, along with impairments in oxidative defense and mitochondrial function. These effects were associated with striatal degeneration, suggesting a neurotoxic impact of HP [4,5,6,7,10,11,12].

Studies have reported a strong association between HP-induced OD and the presence of nitrosative and oxidative stress in the striatum [5,6,7,11,12]. In our current study, we observed similar results, demonstrating that HP administration led to increased levels of nitrite and TBARS (a marker of lipid peroxidation), as well as decreased levels of GSH (an endogenous antioxidant), SOD, CAT (cellular antioxidant enzymes), SDH, ATPase, and ETC enzymes in the rat striatum. These findings support the involvement of NO/free radical toxicity and mitochondrial dysfunction in HP-induced OD. HP belongs to the butyrophenone class of neuroleptics and acts as a DA antagonist by blocking D2 receptors, leading to supersensitivity in the postsynaptic striatal dopamine D2 receptor, resulting in increased turnover of DA. The accelerated DA turnover promotes the generation of reactive metabolites and hydrogen peroxide, leading to elevated oxidative stress specifically in dopaminergic neurons [5,6,7,12,39,40]. Additionally, the autoxidation of DA can generate o-quinone aminochrome, which can further lose an electron to form a leukoaminochrome o-semiquinone radical. This radical has been identified as a significant source of endogenous reactive species. Moreover, the increased DA turnover, coupled with enhanced glutamatergic transmission after HP treatment, leading to striatal excitotoxicity, further amplifies the generation of free radicals and oxidative stress and upregulates corresponding to hyperkinetic dyskinetic movements originating from the striatum [12,39,40]. This study highlights the BT was able to diminish the HP-induced increased level of NO as well as LPO and upregulate the levels of GSH, SOD, and CAT that were found reduced in the rat striatum, suggesting that BT could directly reduce oxidative stress levels and produce antioxidant action as well as indirectly modulate other antioxidant enzyme activities. In addition, BT was able to prevent HP-induced OD; these results may underlie, at least in part, the anti-OD developed property of BT by decreasing excessive NO/free radicals and increasing anti-oxidation power in rat striata.

By inhibiting complex-I (NADH: ubiquinone oxidoreductase) of the electron transport chain in the inner mitochondrial membrane, HP disrupts mitochondrial respiration and enhances the production of reactive oxygen species (ROS) [28,40,41]. Consequently, mitochondrial dysfunction activates N-methyl-D-aspartate (NMDA) receptors, leading to further impairment of mitochondrial function [42]. The hyperexcitability resulting from excessive NMDA receptor stimulation causes an influx of calcium ions (Ca^2+^) and the generation of ROS and reactive nitrogen species (RNS), which induce lipid peroxidation, mitochondrial damage, and DNA damage in both mitochondria and the nucleus [42,43]. The reciprocal interaction between mitochondria and NMDA receptors amplifies the state of nitrative and oxidative stress provoked by HP. Additionally, under pathological conditions associated with increased production of ROS and RNS, nitric oxide (NO) inhibits crucial enzymes involved in energy metabolism and becomes detrimental, causing further damage to cellular components [11,12,44]. In light of these findings, it is evident that both nitrative and oxidative stress, as well as mitochondrial dysfunction, play significant roles in HP-induced OD. Excessive nitrosative and oxidative stress triggers an inflammatory response and the release of inflammatory mediators such as TNF-α, IL-1β, and IL-6, initiating the apoptotic pathway. This pathway has been proposed as a key mechanism for neuronal cell death and a critical pathway in the progression of OD [4,6,10,12,45]. Consistent with previous research, we observed elevated levels of TNF-α, IL-1β, and IL-6 in the striatum of rats treated with HP, indicating the involvement of neuroinflammatory cascades in HP-induced neurotoxicity and the development of OD. Furthermore, caspase-3, known for its involvement in various apoptotic processes [4,12,45], was found to be increased after HP treatment in our study, supporting the role of the apoptotic cascade in the pathology of OD in the rat striatum. This finding aligns with previous preclinical studies demonstrating that animal models of OD induced by prolonged exposure to HP exhibit altered neuronal activity in the striatum due to neuronal injury or death [4,8,12,45]. In this study, BT was able to reduce HP-induced elevated levels of TNF-α, IL-1β, IL-6, and caspase-3, while simultaneously increasing the levels of SDH, ATPase, and ETC enzymes in the rat striatum. Our findings also suggest that BT could exert its neuroprotective effecting by maintaining mitochondrial function and suppressing inflammation and apoptotic cascades in the striatum.

BT, as reported in previous studies, possesses potent antioxidative, anti-inflammatory, and antiapoptotic effects, and it can also help prevent neurochemical deficiencies [13,15,17,18,19,20,21,22,24,26,27]. In our study, we found that BT exhibits multi-functional properties that are associated with various pathophysiological pathways and function against HP-induced OD, as demonstrated by our behavioral assessments. Previous research has also highlighted the protective role of GSH, SOD, CAT, SDH, ATPase, and ETC enzymes in inhibiting the development of OD induced by HP [4,5,6,7,8,9,10,11,12,28]. Therefore, the beneficial effects of BT observed in HP-treated rats can be attributed, at least in part, to its ability to reduce excessive nitric oxide (NO) and free radicals, enhance antioxidative factors, maintain mitochondrial function, and suppress inflammation and apoptotic cascades in the striatum. However, further cellular and molecular studies will be required to confirm the possible neuroprotective mechanisms of BT.

## 5. Conclusions

In conclusion, our study provides compelling evidence for the therapeutic potential of BT in the treatment of OD using an animal model. The results suggest that the neuroprotective effects of BT involve mechanisms such as antioxidation, prevention of mitochondrial dysfunction, attenuation of neuroinflammation, and inhibition of apoptotic cascades. As HP-induced OD in rats serves as a reliable model for studying the neuropathology of TD, our findings lay a foundation for future clinical investigations to assess the efficacy of BT as an adjunct therapy, addressing the safety concerns associated with TD. However, further research is necessary to bridge the gap between preclinical findings and clinical studies in order to gain a comprehensive understanding of the role of BT.

## Figures and Tables

**Figure 1 brainsci-13-01064-f001:**
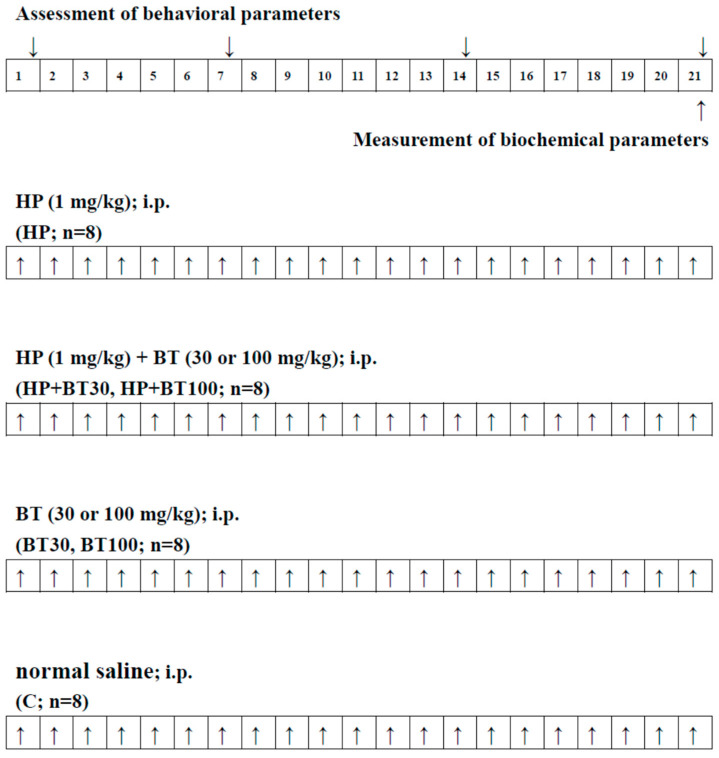
Experimental design and drug treatment paradigm. ↑↓: The day of treatment or assessment.

**Figure 2 brainsci-13-01064-f002:**
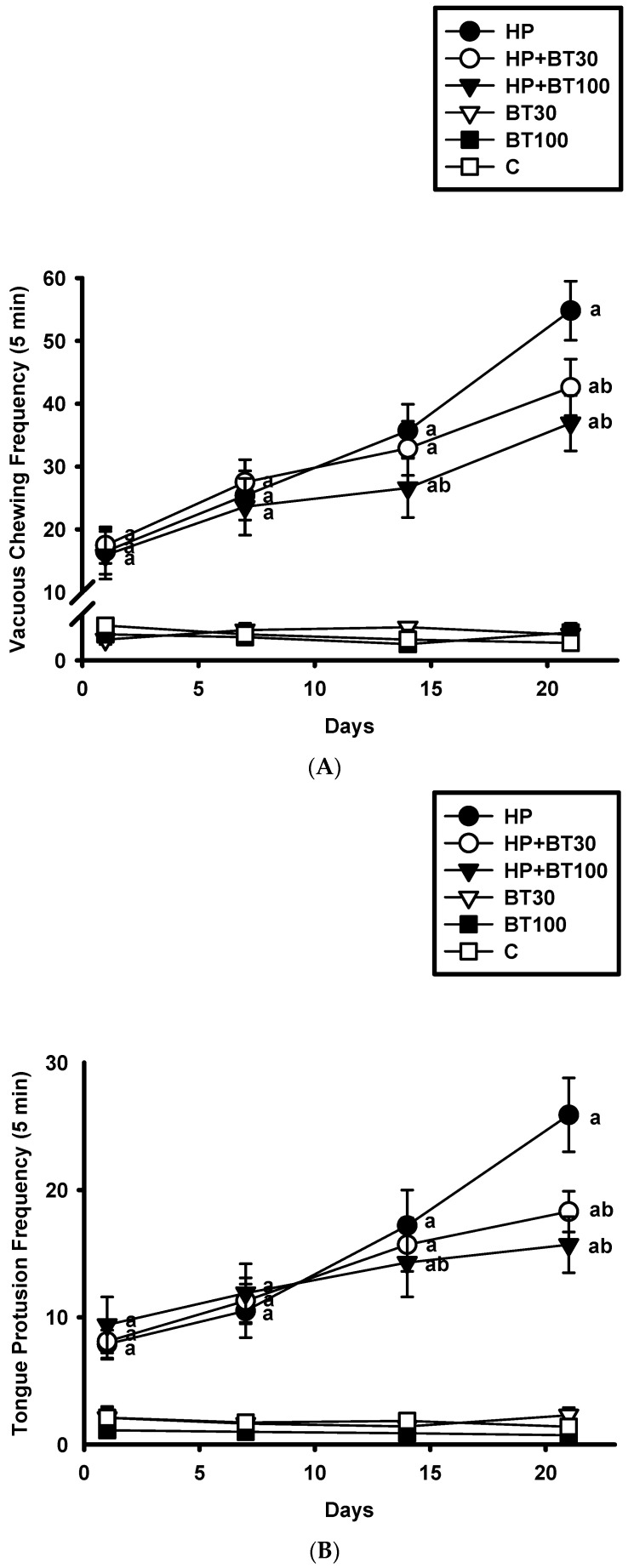
The effects of BT on the HP-induced orofacial dyskinesia (OD) behaviors; (**A**) vacuous chewing movement (VCM) and (**B**) tongue protrusion (TP) in rats. Data are presented as mean ± SEM (n = 8). Two-way ANOVA with Tukey’s test: a *p* < 0.001 as compared with C; b *p* < 0.001 as compared with HP.

**Figure 3 brainsci-13-01064-f003:**
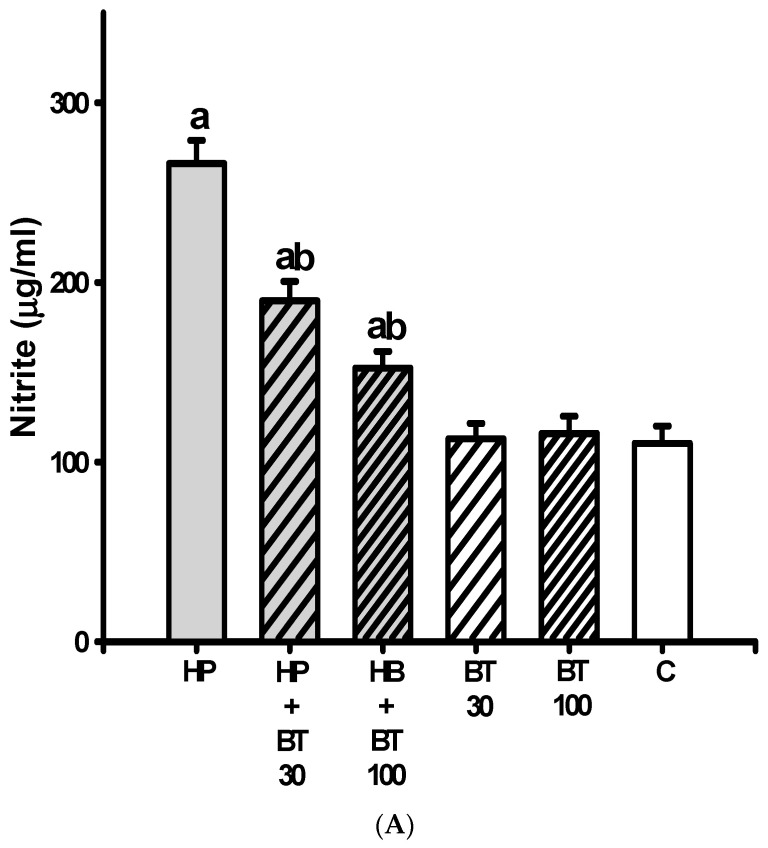
The effects of BT on HP-induced striatal nitrosative and oxidative stress; (**A**) nitrite and (**B**) thiobarbituric acid-reactive substance (TBARS) in rats. Data are presented as mean ± SEM (n = 8). One-way ANOVA with Tukey’s test: a *p* < 0.001 as compared with C; b *p* < 0.001 as compared with HP.

**Figure 4 brainsci-13-01064-f004:**
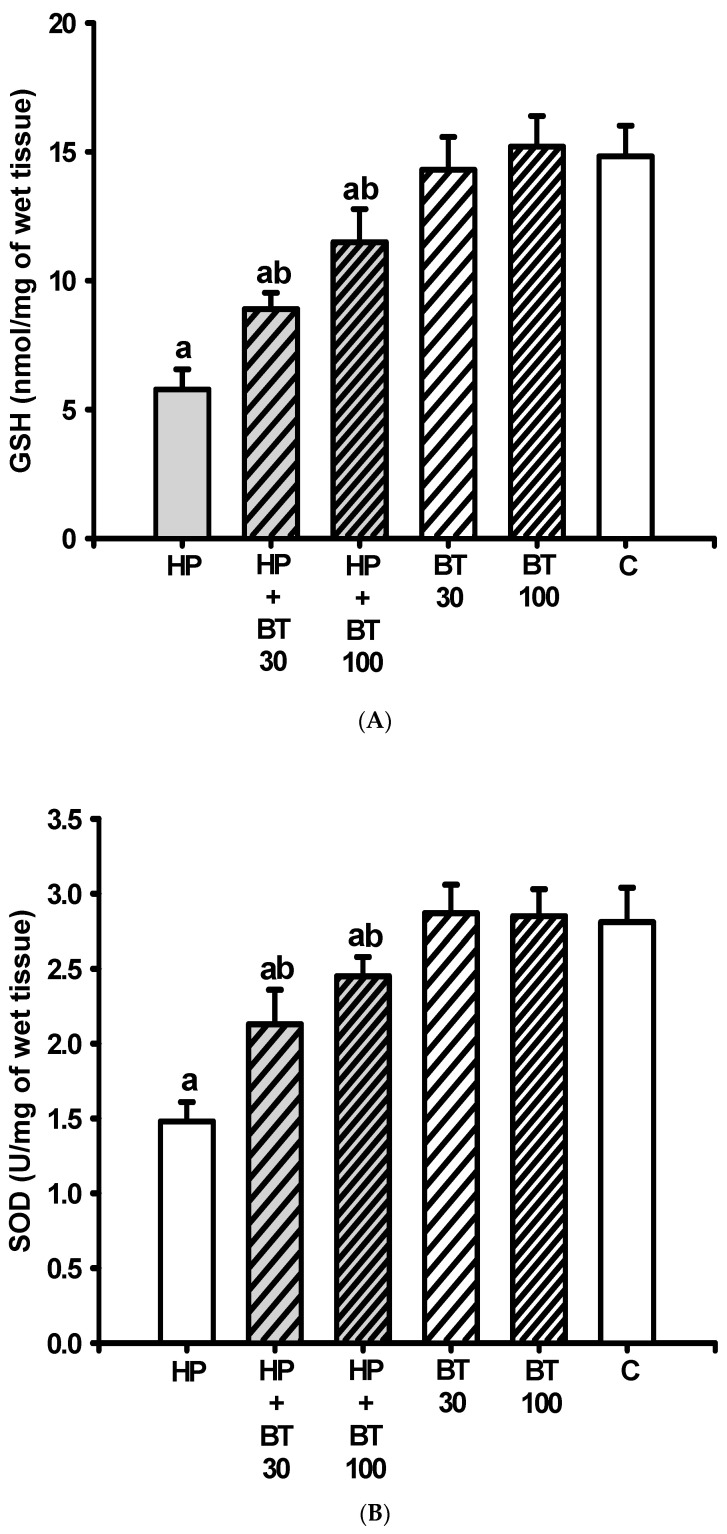
The effects of BT on HP-induced decreased striatal antioxidation power; (**A**) glutathione (GSH), (**B**) superoxide dismutase (SOD), and (**C**) catalase (CAT) in rats. Data are presented as mean ± SEM (n = 8). One-way ANOVA with Tukey’s test: a *p* < 0.001 as compared with C; b *p* < 0.001 as compared with HP.

**Figure 5 brainsci-13-01064-f005:**
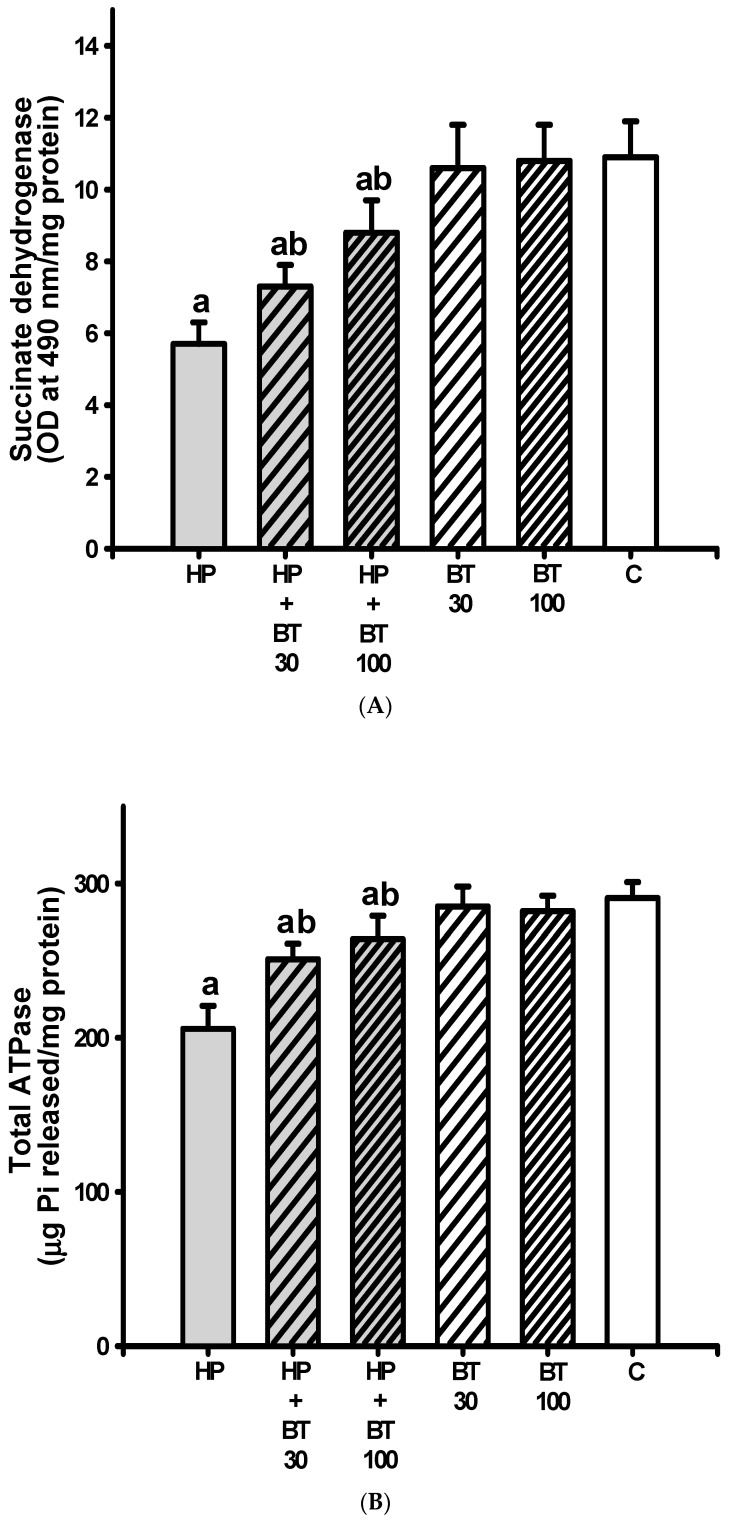
The effects of BT on HP-induced striatal mitochondrial dysfunction; (**A**) SDH, (**B**) total ATPase, (**C**) NADH-cytochrome C reductase (complex I–III), and (**D**) succinate-cytochrome C reductase (complex II–III) in rats. Data are presented as mean ± SEM (n = 8). One-way ANOVA with Tukey’s test: a *p* < 0.001 as compared with C; b *p* < 0.001 as compared with HP.

**Figure 6 brainsci-13-01064-f006:**
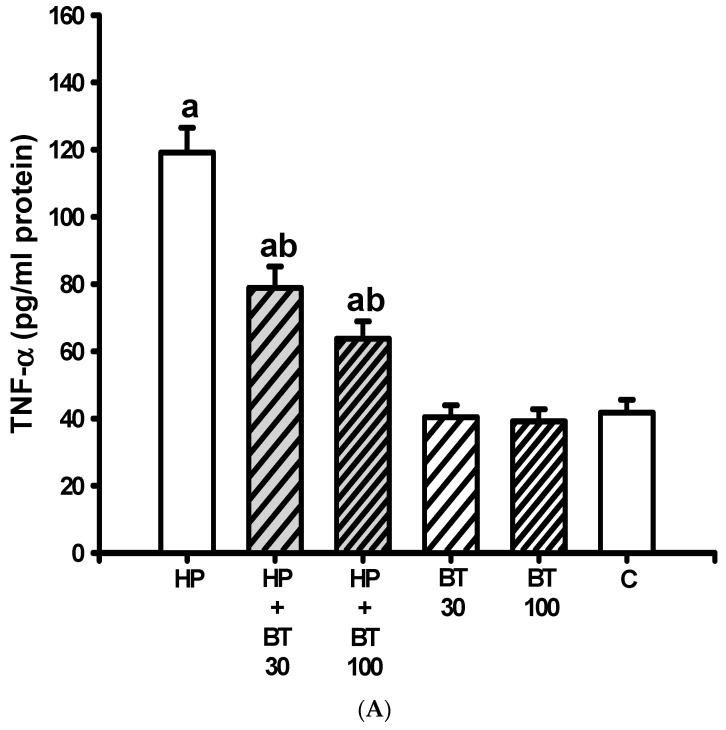
The effects of BT on HP-induced increased striatal neuroinflammatory and apoptotic markers; (**A**) tumor necrosis factor α (TNF-α), (**B**) interleukin-1β (IL-1β), (**C**) interleukin-6 (IL-6), and (**D**) Caspase-3 in rats. Data are presented as mean ± SEM (n = 8). One-way ANOVA with Tukey’s test: a *p* < 0.001 as compared with C; b *p* < 0.001 as compared with HP.

## Data Availability

Not applicable.

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
