# Peer review of "Involvement of Antioxidant and Prevention of Mitochondrial Dysfunction, Anti-Neuroinflammatory Effect and Anti-Apoptotic Effect: Betaine Ameliorates Haloperidol-Induced Orofacial Dyskinesia in Rats"

_brainsci, 2023, doi:10.3390/brainsci13071064_

Round 1
Reviewer 1 Report
1. In material and method: The author should explain why they selected only male Wistar rats for the study. Also, discuss the exclusion criteria for female rats.
2. The study includes three months old rats and mentioned a weighing range between 270 and 320 g, why author used high-weight animals?
3. The author should explain why they include two perse groups instead of one perse group (Betaine 30 mg/kg and 100 mg/kg), generally, the researcher utilizes the higher dose perse group.
4. The solvent utilized to dissolve the drugs should be discussed in the material and method section.
5. In Figure 1 of the experimental design and drug treatment paradigm, the author wrote that behaviour parameters were assessed on the 1st, 7th, 14th, and 21st, Whereas, figure 2a and 2b showed that VCMS and Tongue protrusion was performed on the day 0, 5, 10, 15, and 20. It should be the same in both places.
6. In figure 2: Three-way ANOVA was used to assessed the VCMs and TP but we didn’t find more than two variables in this figure, it should be Two-Way or if any exception author should justify.
7. For biochemical analysis, the author used Two-way ANOVA which is not correct statistically, because for Two-way ANOVA you should have at least two variables but for biochemical, you have only one variable.
8. In discussions author wrote “Based on our current understanding, our findings reveal for the initial time that BT exhibits robust protective properties against HP-induced pathophysiological dysfunction, suggesting its potential therapeutic value in treating TD in humans. Please rephrase it
9. 3.5. Statistical analysis should be re-write.
10. The mechanism of action of Betaine was not explored in this study it should be better if the author should be performed Western and if not possible then explain the mechanism and linked it with current parameters in the discussion
Zhang, Man et al. “Betaine Inhibits NLRP3 Inflammasome Hyperactivation and Regulates Microglial M1/M2 Phenotypic Differentiation, Thereby Attenuating Lipopolysaccharide-Induced Depression-Like Behavior.” Journal of immunology research vol. 2022 9313436. 26 Oct. 2022, doi:10.1155/2022/9313436
11. In Conclusion: The author should explain the limitation of the study and discuss the future prospects for other neurological disorders.
12. There are typos, grammatical/style errors, and incorrect terminology used throughout the manuscripts, indicating a need for critical revision.
There are typos, grammatical/style errors, and incorrect terminology used throughout the manuscripts, indicating a need for critical revision.
Author Response
Response to Reviewer 1 Comments
Dear Reviewers,
I’m writing in response to your feedback regarding the manuscript we submitted. Thank you so much for your positive comments and suggestions for the manuscript entitled. The manuscript has been revised based on your inquiries, rephrased the content of the manuscript and resubmitted through the journal website. The revised parts have been marked in red. The following is the response to your inquiries point-by-point; we hope our responses fully address your comments and suggestions:
Reviewers’ comments:
Comments and Suggestions for Authors
Point 1: In material and method: The author should explain why they selected only male Wistar rats for the study. Also, discuss the exclusion criteria for female rats.
Response 1:
Thank you so much for giving us such detail constructive comments and suggestions. In order to exclude that the menstrual cycle of female rats may affect the experimental results, male rats were selected as experimental animals in this experiment. Female rats will be considered for the next study.
Point 2: The study includes three months old rats and mentioned a weighing range between 270 and 320 g, why author used high-weight animals?
Response 2:
Thank you so much for giving us such detail suggestions, in this study, rats with high body weight were not deliberately selected as experimental subjects, just because three-month-old rats weighing exactly between 270-320 g.
Point 3: The author should explain why they include two perse groups instead of one perse group (Betaine 30 mg/kg and 100 mg/kg), generally, the researcher utilizes the higher dose perse group.
Response 3:
Thank you again for the constructive suggestions. In the present study, the current choice of dosage was based on previous study (Chen et al., 2021) and our previous laboratory findings in order to know the minimal and maximal efficacy of betaine (BT) in this model. BT was administrated intraperitoneally and pretested from the low dose of 1 mg/kg, owing to no evident effect for the low dosage application; the dose was increased gradually and eventually up to 300 mg/kg to observe the statistically significant effect for BT treatment. Therefore the 30/100 mg/ kg dosage were adapted to the current study.
References:
Chen ST, Hsieh CP, Lee MY, Chen LC, Huang CM, Chen HH, Chan MH. Betaine prevents and reverses the behavioral deficits and synaptic dysfunction induced by repeated ketamine exposure in mice. Biomed Pharmacother. 2021 Dec;144:112369.
Point 4: The solvent utilized to dissolve the drugs should be discussed in the material and method section.
Response 4:
Thank you so much for giving us such detail suggestions, we have added the composition of solvent in the revised manuscript.
Point 5: In Figure 1 of the experimental design and drug treatment paradigm, the author wrote that behaviour parameters were assessed on the 1st, 7th, 14th, and 21st, Whereas, figure 2a and 2b showed that VCMS and Tongue protrusion was performed on the day 0, 5, 10, 15, and 20. It should be the same in both places.
Response 5:
Thank you so much for giving us such a detail constructive comments and suggestions, In Figure 1 of the experimental design and drug treatment paradigm, behaviour parameters were assessed on the 1st, 7th, 14th, and 21st day, Because the drawing software cannot correct the interval of the X-axis, figure 2a and 2b showed that VCMS and Tongue protrusion was look like performed on the day 0, 5, 10, 15, and 20. Figure 2a and 2b have been redrawn for greater clarity in the revised manuscript. The drawing software will be upgraded as soon as possible.
Point 6: In figure 2: Three-way ANOVA was used to assessed the VCMs and TP but we didn’t find more than two variables in this figure, it should be Two-Way or if any exception author should justify.
Point 7: For biochemical analysis, the author used Two-way ANOVA which is not correct statistically, because for Two-way ANOVA you should have at least two variables but for biochemical, you have only one variable.
Response 6 and 7:
Thank you so much for giving us such a detail constructive comments and suggestions again, we have corrected thesein the revised manuscript.
Point 8: In discussions author wrote “Based on our current understanding, our findings reveal for the initial time that BT exhibits robust protective properties against HP-induced pathophysiological dysfunction, suggesting its potential therapeutic value in treating TD in humans. Please rephrase it
Response 8:
Thank you so much again, we have rephrased these in the revised manuscript.
Point 9: 3.5. Statistical analysis should be re-write.
Response 9:
Thank you so much for giving us such a detail constructive comments and suggestions, Statistical analysis has beenrewritten in the revised manuscript.
Point10: The mechanism of action of Betaine was not explored in this study it should be better if the author should be performed Western and if not possible then explain the mechanism and linked it with current parameters in the discussion
Zhang, Man et al. “Betaine Inhibits NLRP3 Inflammasome Hyperactivation and Regulates Microglial M1/M2 Phenotypic Differentiation, Thereby Attenuating Lipopolysaccharide-Induced Depression-Like Behavior.” Journal of immunology research vol. 2022 9313436. 26 Oct. 2022, doi:10.1155/2022/9313436
Response 10:
Thank you so much again, the mechanism of action of betaine was been discussed in this study accordingly and this report was been cited in this study.
Point 11: In Conclusion: The author should explain the limitation of the study and discuss the future prospects for other neurological disorders.
Response 11:
Thank you so much for the suggestion again, we have explained the limitation of the study and discussed the future prospects for other neurological disorders in the revised manuscript.
Point 12: There are typos, grammatical/style errors, and incorrect terminology used throughout the manuscripts, indicating a need for critical revision.
Response 12:
Thank you again for the comments and suggestions. The manuscript has been rerevised and corrected. Before we finalized the revised manuscript, we have resent the manuscript for proof-reading as well as one last review by a native English-speaking researcher.
Thank you for your valuable comments/suggestions and giving us the opportunity to revise the manuscript to a more readable level. We worked very hard to response your inquiries and to revise the manuscript. We hope the manuscript could pass the review to be published in your
prestigious journal: Brain Sciences
Sincerely yours,
Hung-Sheng Soung
Department of Psychiatry, Yuan-Shan Br. of Taipei Veteran General Hospital, No.386, Rongguang Rd., Neicheng, Yuanshan Township, Yilan County 264, Taiwan.
Fax: +886 3 922 2141.
E-mail address: KGC5874@gmail.com (H-S Soung).

Reviewer 2 Report
In my point of view, a major revision is indicated, which could improve the quality and understanding of the manuscript.
Title:
1)The title is not expressing well the founds of the study, the authors should add the neuroprotective mechanisms involved.
Abstract:
2) The authors should add at the beginning of the abstract why it is important and/or clinically relevant to the study of haloperidol-induced dyskinesia.
3) It is not clear that the treatment with betaine is concomitant with the injections of haloperidol, the authors must delete words like “following” HP injection.
4) In the last phrase of the abstract the authors relate betaine as a novel therapeutic candidate in delaying or treating human TD in clinical settings. The authors must not do this direct suggestion, it is important to highlight the need for future clinical studies.
Keywords:
5) The authors could exchange the words for different ones from the title to expand the search for the article.
Introduction:
6) The authors must clarify that haloperidol promotes side effects such as tardive dyskinesia and that is why the authors did this model of experiment.
7) In the first paragraph the authors mentioned that tardive dyskinesia is “characterized by involuntary movements, such as choreiform, athetoid, and rhythmic motions, predominantly involving the mouth, face, and tongue”. If TD presents choreiform movements the authors must add a reference at the end of the sentence.
8) The authors should clarify that orofacial dyskinesia is observed in rats only and is mimicking TD observed in humans.
9) It would be interesting if the authors of the manuscript mentioning about the model of the protocol and make the connection with the clinic and TD in the last paragraph of the introduction, as a justification of the study together with the hypothesis of the study.
10) Considering the objectives and the findings of the study, in the second paragraph the authors should add more information about studies of the benefits of betaine regarding only the finds in the central nervous system, it is not necessary the information about betaine in other systems. Also, information like “BT plays a crucial role in one-carbon metabolism, acting as an essential methyl group donor in transmethylation reactions.” Are confusing and distracting and are not important for the study.
11) The authors should add in the second paragraph where is found betaine and what food sources are.
12) The authors must add a reference in the sentence “It is subsequently distributed to different organs, including the brain.”. It is important to know that betaine can pass the blood-brain barrier.
13) The authors should delete any methods contained in the third paragraph.
Materials and Methods:
14) In the introduction section the authors explain that betaine is a “safe dietary nutrient
supplement” If so, why the authors chose to do the administration of betaine intraperitoneally? The authors must clarify this in the methods section.
15) In item 2.3 of materials and methods the authors should clarify if the groups that do not receive haloperidol received vehicle solution as mentioned in item 2.4. The same thing in Figure number 1, it is not clear that the animals received vehicle solutions in any group.
16) In item 2.4 the authors mentioned that the behavioral assessment was made after six hours of the injection of haloperidol, since the animals received the injection of betaine one hour after haloperidol the authors could express that the behavioral assessment was made after haloperidol and betaine injection.
17) The authors mentioned in item 2.5 that “Their brains were swiftly removed and rinsed with ice-cold saline solution to eliminate any residual blood. Subsequently, the brains were stored at a temperature of -80°C. The striatum, a specific brain region, was carefully dissected from the intact brain on an ice plate, following the stereotaxic atlas provided by Budantsev et al. [21].”. Usually, all researchers of the world do not freeze the whole brain for a posterior removal of one region, if the authors did this, must add a reference to this method or clarify that the striatum was removed immediately after the euthanasia.
18) In Figure 2a it is not possible to see group B100, the authors should fix that.
19) Why the authors did not do the test of catalepsy? It is mentioned in the introduction that haloperidol promotes this behavior.
20) In the whole manuscript the authors mentioned betaine as “BT” but in the figures is only B and the same thing with haloperidol as ‘HP’ and in the figures is only H. The authors must standardize and use the same all over the manuscript.
Discussion:
21) The authors must not do a direct suggestion of TD with the results found in this study, it is important to highlight the need for future clinical studies as mentioned in the conclusion section.
22) The authors should add a reference in line 415 after the phrase “Our findings regarding OD induced by HP align with those of other researchers and our own previous studies” and cite other studies from the group and other researchers.
23) The biggest finding of the study is that betaine decreases oxidative stress levels, increased anti-oxidation levels, prevented mitochondrial dysfunction, and reduced the levels of neuroinflammatory and apoptotic markers and this is not well explored by the authors in the discussion section. The authors must explore the mechanism involving haloperidol-induced oxidative stress and clarify how all of the results found in this study are correlated with that, also pointing to the pre and post-synaptic effect of haloperidol. The authors should not discuss the founds of the study as a separate process of the brain and highlight the benefits of the use of betaine.
24)The last paragraph mentions the need to explore new therapeutic approaches for TD in medical research until line 473, this should not be in the discussion section, it is important to justify the study in the introduction section.
25) The authors mentioned that anticholinergic drugs such as biperiden are used commonly with antipsychotic drugs. Although this could happen, the authors must know that there is no treatment for TD available in the clinical practice and must delete all sentences that mention biperiden as a bad treatment for TD. New therapeutic approaches for TD must be researched because there is no one available.
Conclusion:
26) Terms like compelling evidence and strongly suggest must be removed from the manuscript. The founds of the study should be a suggestion as a potential adjunct therapy along with more pre-clinical and clinical studies.
No comments
Author Response
Response to Reviewer 2 Comments
Dear Reviewers,
I’m writing in response to your feedback regarding the manuscript we submitted. Thank you so much for your positive comments and suggestions for the manuscript entitled. The manuscript has been revised based on your inquiries, rephrased the content of the manuscript and resubmitted through the journal website. The revised parts have been marked in red. The following is the response to your inquiries point-by-point; we hope our responses fully address your comments and suggestions:
Reviewers’ comments:
Comments and Suggestions for Authors
In my point of view, a major revision is indicated, which could improve the quality and understanding of the manuscript.
Title:
Point 1: The title is not expressing well the founds of the study, the authors should add the neuroprotective mechanisms involved.
Response 1: Thank you so much for giving us such detail constructive comments and suggestions, the neuroprotective mechanisms involved in the title in the revised manuscript.
Abstract:
Point 2: The authors should add at the beginning of the abstract why it is important and/or clinically relevant to the study of haloperidol-induced dyskinesia.
Response 2: Thank you so much for giving us such detail constructive suggestions, the important and/or clinically relevant to the study of haloperidol-induced dyskinesia has been added at the beginning of the abstract in the revised manuscript.
Point 3: It is not clear that the treatment with betaine is concomitant with the injections of haloperidol, the authors must delete words like “following” HP injection.
Response 3: Thank you so much again, the words “following” HP injection has been deleted in the revised manuscript.
Point 4: In the last phrase of the abstract the authors relate betaine as a novel therapeutic candidate in delaying or treating human TD in clinical settings. The authors must not do this direct suggestion, it is important to highlight the need for future clinical studies.
Response 4: Thank you so much for giving us such detail constructive comments and suggestions, we have corrected these in the revised manuscript.
Keywords:
Point 5: The authors could exchange the words for different ones from the title to expand the search for the article.
Response 5: Thank you so much for such detail constructive suggestions, the keywords have been exchanged to expand the search for the article in the revised manuscript.
Introduction:
Point 6: The authors must clarify that haloperidol promotes side effects such as tardive dyskinesia and that is why the authors did this model of experiment.
Response 6: Thank you so much again, we have clarified that haloperidol promote side effects and that is why we did this animal model of experiment in the revised manuscript.
Point 7: In the first paragraph the authors mentioned that tardive dyskinesia is “characterized by involuntary movements, such as choreiform, athetoid, and rhythmic motions, predominantly involving the mouth, face, and tongue”. If TD presents choreiform movements the authors must add a reference at the end of the sentence.
Response 7: Thank you so much, we have added a reference accordingly in the revised manuscript.
Point 8: The authors should clarify that orofacial dyskinesia is observed in rats only and is mimicking TD observed in humans.
Response 8: Thank you so much for such detail constructive suggestions, we have clarified that orofacial dyskinesia is observed in rats only and is mimicking TD observed in humans in the revised manuscript.
Point 9: It would be interesting if the authors of the manuscript mentioning about the model of the protocol and make the connection with the clinic and TD in the last paragraph of the introduction, as a justification of the study together with the hypothesis of the study.
Response 9: Thank you so much for such detail constructive suggestions, we have mentioned about the model of the protocol and make the connection with the clinic and TD in the last paragraph of the introduction in the revised manuscript.
Point 10: Considering the objectives and the findings of the study, in the second paragraph the authors should add more information about studies of the benefits of betaine regarding only the finds in the central nervous system, it is not necessary the information about betaine in other systems. Also, information like “BT plays a crucial role in one-carbon metabolism, acting as an essential methyl group donor in transmethylation reactions.” Are confusing and distracting and are not important for the study.
Response 10: Thank you so much for such detail constructive suggestions, we have added more information about studies of the benefits of BT in the central nervous system in the revised manuscript.
Point 11: The authors should add in the second paragraph where is found betaine and what food sources are.
Response 11: Thank you so much for such detail constructive suggestions, we have added where is found betaine and what food sources are in the second paragraph in the revised manuscript.
Point 12: The authors must add a reference in the sentence “It is subsequently distributed to different organs, including the brain.”. It is important to know that betaine can pass the blood-brain barrier.
Response 12: Thank you so much for suggestions, we have added a reference in the revised manuscript.
Point 13: The authors should delete any methods contained in the third paragraph.
Response 13: Thank you so much for such detail constructive suggestions, we have deleted any methods contained in the third paragraph in the revised manuscript.
Materials and Methods:
Point 14: In the introduction section the authors explain that betaine is a “safe dietary nutrient supplement” If so, why the authors chose to do the administration of betaine intraperitoneally? The authors must clarify this in the methods section.
Response 13: Thank you so much for such detail constructive suggestions, in order for the drug concentration to enter the animal body accurately, HP and BT were administered intraperitoneally (i.p.) at a volume of 2.0 mL/kg body weight. The administration of betaine orally will be considered carefully in our ongoing research project.
Point 15: In item 2.3 of materials and methods the authors should clarify if the groups that do not receive haloperidol received vehicle solution as mentioned in item 2.4. The same thing in Figure number 1, it is not clear that the animals received vehicle solutions in any group.
Response 15: Thank you so much for suggestions, we have changed these in the revised manuscript.
Point 16: In item 2.4 the authors mentioned that the behavioral assessment was made after six hours of the injection of haloperidol, since the animals received the injection of betaine one hour after haloperidol the authors could express that the behavioral assessment was made after haloperidol and betaine injection.
Response 16: Thank you so much for suggestions again, we have changed these in the revised manuscript.
Point 17: The authors mentioned in item 2.5 that “Their brains were swiftly removed and rinsed with ice-cold saline solution to eliminate any residual blood. Subsequently, the brains were stored at a temperature of -80°C. The striatum, a specific brain region, was carefully dissected from the intact brain on an ice plate, following the stereotaxic atlas provided by Budantsev et al. [21].”. Usually, all researchers of the world do not freeze the whole brain for a posterior removal of one region, if the authors did this, must add a reference to this method or clarify that the striatum was removed immediately after the euthanasia.
Response 17: Thank you so much for suggestions again, we have corrected these in the revised manuscript.
Point 18: In Figure 2a it is not possible to see group B100, the authors should fix that.
Response 18: Thank you so much for giving us such detail constructive comments and suggestions, we have fixedFigure 2a as best as we can in the revised manuscript.
Point 19: Why the authors did not do the test of catalepsy? It is mentioned in the introduction that haloperidol promotes this behavior.
Response 19: Thank you so much for the suggestion, the test of catalepsy will be considered carefully in our ongoing research project.
Point 20: In the whole manuscript the authors mentioned betaine as “BT” but in the figures is only B and the same thing with haloperidol as ‘HP’ and in the figures is only H. The authors must standardize and use the same all over the manuscript.
Response 20: Thank you so much for such detail constructive suggestions, we have corrected these in the revised manuscript.
Discussion:
Point 21: The authors must not do a direct suggestion of TD with the results found in this study, it is important to highlight the need for future clinical studies as mentioned in the conclusion section.
Response 21: Thank you so much for such detail constructive suggestions, we have corrected these in the revised manuscript.
Point 22: The authors should add a reference in line 415 after the phrase “Our findings regarding OD induced by HP align with those of other researchers and our own previous studies” and cite other studies from the group and other researchers.
Response 22: Thank you so much for such detail constructive suggestions, we have cited references in the revised manuscript.
Point 23: The biggest finding of the study is that betaine decreases oxidative stress levels, increased anti-oxidation levels, prevented mitochondrial dysfunction, and reduced the levels of neuroinflammatory and apoptotic markers and this is not well explored by the authors in the discussion section. The authors must explore the mechanism involving haloperidol-induced oxidative stress and clarify how all of the results found in this study are correlated with that, also pointing to the pre and post-synaptic effect of haloperidol. The authors should not discuss the founds of the study as a separate process of the brain and highlight the benefits of the use of betaine.
Response 23: Thank you again for the comments and suggestions. The discussion section has been rewritten. We have discussed
Point 24: The last paragraph mentions the need to explore new therapeutic approaches for TD in medical research until line 473, this should not be in the discussion section, it is important to justify the study in the introduction section.
Response 24: Thank you so much for suggestions again, we have changed these in the revised manuscript.
Point 25: The authors mentioned that anticholinergic drugs such as biperiden are used commonly with antipsychotic drugs. Although this could happen, the authors must know that there is no treatment for TD available in the clinical practice and must delete all sentences that mention biperiden as a bad treatment for TD. New therapeutic approaches for TD must be researched because there is no one available.
Response 25: Thank you so much for such detail constructive suggestions, we have corrected these in the revised manuscript.
Conclusion:
Point 26: Terms like compelling evidence and strongly suggest must be removed from the manuscript. The founds of the study should be a suggestion as a potential adjunct therapy along with more pre-clinical and clinical studies.
Response 26: Thank you so much for such detail constructive suggestions, we have corrected these in the revised manuscript.
Comments on the Quality of English Language
No comments
The manuscript has been rerevised and corrected. Before we finalized the revised manuscript, we have resent the manuscript for proof-reading as well as one last review by a native English-speaking researcher.
Thank you for your valuable comments/suggestions and giving us the opportunity to revise the manuscript to a more readable level. We worked very hard to response your inquiries and to revise the manuscript. We hope the manuscript could pass the review to be published in your
prestigious journal: Brain Sciences
Sincerely yours,
Hung-Sheng Soung
Department of Psychiatry, Yuan-Shan Br. of Taipei Veteran General Hospital, No.386, Rongguang Rd., Neicheng, Yuanshan Township, Yilan County 264, Taiwan.
Fax: +886 3 922 2141.
E-mail address: KGC5874@gmail.com (H-S Soung).

Reviewer 3 Report
Review for Manuscript- brainsci-2474486
The manuscript by Hsieh Ming Shaung et al., entitled "Betaine ameliorates haloperidol-induced orofacial dyskinesia in rats" is the research report finding that the impact of Betaine (BT) on haloperidol (HP)-induced orofacial dyskinesia (OD) in rats, as well as the underlying neuroprotective mechanisms. The authors reported that the neuroprotective effects of BT against HP-induced OD are credited to its antioxidant, prevention of mitochondrial dysfunction, anti-neuroinflammatory effect, and anti-apoptotic effect, suggesting that BT may be a novel therapeutic candidate in delaying or treating human TD in clinical settings. The manuscript is well-written and the cited references are appropriate. However, some remarks should be taken by authors under consideration before paper publication. The manuscript needs minor revision before its final publication.
Comments:
1. Authors should modify the schematic diagram for the experimental design. It will help the readers understand the experimental time points.
2. Authors should correct the sub-section numbers after 2.9 continued as 2.10, 2.11…. like that in the materials and methods section.
3. Authors should mention the software and version used for the statistical analysis in the materials and method section.
4. In all the graphs, the authors should maintain the consistency for Haloperidol as “HP” and Betaine as “BT”. Authors should change the H to HP and B to BT in all the graphs.
5. Minor editing is needed throughout the manuscript, especially in the Discussion.
Author Response
Response to Reviewer 3 Comments
Dear Reviewers,
I’m writing in response to your feedback regarding the manuscript we submitted. Thank you so much for your positive comments and suggestions for the manuscript entitled. The manuscript has been revised based on your inquiries, rephrased the content of the manuscript and resubmitted through the journal website. The revised parts have been marked in red. The following is the response to your inquiries point-by-point; we hope our responses fully address your comments and suggestions:
Reviewers’ comments:
Comments and Suggestions for Authors
The manuscript by Hsieh Ming Shaung et al., entitled "Betaine ameliorates haloperidol-induced orofacial dyskinesia in rats" is the research report finding that the impact of Betaine (BT) on haloperidol (HP)-induced orofacial dyskinesia (OD) in rats, as well as the underlying neuroprotective mechanisms. The authors reported that the neuroprotective effects of BT against HP-induced OD are credited to its antioxidant, prevention of mitochondrial dysfunction, anti-neuroinflammatory effect, and anti-apoptotic effect, suggesting that BT may be a novel therapeutic candidate in delaying or treating human TD in clinical settings. The manuscript is well-written and the cited references are appropriate. However, some remarks should be taken by authors under consideration before paper publication. The manuscript needs minor revision before its final publication.
Response: Thank you so much for the kind comment and wonderful suggestion.
Point 1: Authors should modify the schematic diagram for the experimental design. It will help the readers understand the experimental time points.
Response 1: Thank you for the comments and suggestions, we have modified the schematic diagram for the experimental design in the revised manuscript.
Point 2: Authors should correct the sub-section numbers after 2.9 continued as 2.10, 2.11…. like that in the materials and methods section.
Response 2: Thank you for the comments and suggestions, we have corrected sub-section numbers after 2.9 in the revised manuscript.
Point 3: Authors should mention the software and version used for the statistical analysis in the materials and method section.
Response 3: Thank you so much for the suggestion, we have mentioned the software and version used for the statistical analysis in the materials and method section in the revised manuscript.
Point 4: In all the graphs, the authors should maintain the consistency for Haloperidol as “HP” and Betaine as “BT”. Authors should change the H to HP and B to BT in all the graphs.
Response 4: Thank you so much again, we have checked again and corrected these in the revised manuscript.
Point 5: Minor editing is needed throughout the manuscript, especially in the Discussion.
Response 5: Thank you again for the comments and suggestions. The manuscript been re-edited, the discussion section has been rewritten. Before we finalized the revised manuscript, we have resent the manuscript for proof-reading as well as one last review by a native English-speaking researcher.
Thank you for your valuable comments/suggestions and giving us the opportunity to revise the manuscript to a more readable level. We worked very hard to response your inquiries and to revise the manuscript. We hope the manuscript could pass the review to be published in your
prestigious journal: Brain Sciences
Sincerely yours,
Hung-Sheng Soung
Department of Psychiatry, Yuan-Shan Br. of Taipei Veteran General Hospital, No.386, Rongguang Rd., Neicheng, Yuanshan Township, Yilan County 264, Taiwan.
Fax: +886 3 922 2141.
E-mail address: KGC5874@gmail.com (H-S Soung).

Round 2
Reviewer 2 Report
The Authors made the necessary changes, as required.